# Adaptive Points Sampling for Implicit Field Reconstruction of Industrial Digital Twin

**DOI:** 10.3390/s22176630

**Published:** 2022-09-02

**Authors:** Jiongchao Jin, Huanqiang Xu, Biao Leng

**Affiliations:** 1School of Computer Science and Engineering, Beihang University, Beijing 100191, China; 2Beijing GlacierAI Technology Co., Ltd., Beijing 100084, China

**Keywords:** deep learning, digital twin, implicit field, 3D reconstruction

## Abstract

Nowadays, the digital twin (DT) plays an important role in Industry 4.0. It aims to model reality in the digital space for further industrial maintenance, management, and optimization. Previously, many AI technologies have been applied in this field and provide strong tools to connect physical and virtual spaces. However, we found that single-view 3D reconstruction (SVR) for DT has not been thoroughly studied. SVR can generate 3D digital models of real industrial products from just a single image. The application of SVR technology would bring convenience, cheapness, and robustness to modeling physical objects in digital space. However, the existing SVR methods cannot perform well in the reconstruction of details, which is indispensable and challenging in industrial products. In this paper, we propose a new detail-aware feature extraction network based on a feature pyramid network (FPN) for better detail reconstruction. Then, an extra network is designed to combine convolutional feature maps from different levels. Moreover, we also propose a novel adaptive points-sampling strategy to adaptively change the learning difficulty according to the training status. This can accelerate the training process and improve the fine-tuned network performance as well. Finally, we conduct comprehensive experiments on both the general objects dataset ShapeNet and a collected industrial dataset to prove the effectiveness of our methods and the practicability of the SVR technology for DT.

## 1. Introduction

With the rapid development of the Internet of things (IoT) in Industry 4.0, the interaction between physical and digital spaces has become an essential process in the industrial world. The digital twin (DT) [1,2,3] was brought out to bidirectionally bridge the virtual world and reality. DT is a complex concept, closely related to robotics technology, computer vision, computer graphics, artificial intelligence, and other areas in computer science. It enables real-time modeling, monitoring, analysis, prediction, and control of physical objects [4,5]. DT can also significantly improve industry chain collaboration, urban management, and industrial system optimization [6,7,8,9]. Importantly, the tasks of the digital twin cannot be accomplished without 3D models.

The 3D digital model plays an indispensable role throughout the different stages of industrial production, including industrial product design, manufacturing, and maintenance. However, the interaction between authentic products and 3D digital models is still a challenging problem. It is also an important research topic in DT. In existing systems, sophisticated sensors are needed to capture the 3D data of products to reconstruct 3D digital models. In our paper, we try to utilize single-view 3D reconstruction (SVR) technology to reconstruct 3D models from one 2D image of the real world. Compared with complex sensor-based model reconstruction, an image input can be easily obtained by mobile devices and processed with less computing power, which enables SVR to be applied in a wide range of scenarios.

Nowadays, the most popular SVR methods are based on implicit field [10,11,12]. Previous works have used meshes, point clouds, or voxels to discretely represent a 3D model, which makes it impossible for them to reconstruct a smooth surface for 3D model shapes. However, implicit-field-based SVR methods represent 3D models with a continuous SDF function. In the 3D decoder part, they train a neural network as a binary classifier with supervision by implicit fuction (i.e., SDF function). The decoder network takes the coordinates of 3D points as queries and outputs the predictions of whether the given points are inside or outside of the 3D object’s surface. In this way, the network generates an implicit representation of the 3D objects. When the number of input points is large enough, the surface of the 3D object can be well reconstructed. Implicit representation avoids using neural networks to predict complex geometry structures, which is difficult to do. It also utilizes networks to fit the decision surface, which is an advantage of neural networks. In addition, implicit representation has no limit in resolution. Theoretically, the more query points we input, the more precise of a surface we generate. However, the existing implicit-field-based methods are not suitable for industrial production. Without sufficient utilization of image information and a careful data-sampling strategy, learning with implicit field cannot perform well on sharp edges or surface details [13], which are indispensable for industrial product reconstruction.

To better reconstruct the details of industrial products, the local information in the input image should be utilized. Most of the popular implicit-field-based methods [10,11,12] only use a convolutional neural network (CNN) to extract a single feature vector from the input image, ignoring the local information. Other methods, such as [14,15], obtain multi-level feature maps by CNN and obtain local information from a query point on these feature maps. However, the low-level feature maps are extracted by a tiny network and contain limited semantic information. To handle this problem, we propose a feature-extracting network based on feature pyramid networks (FPN) [16]. Then, an extra network is used to combine high-level feature maps with rich semantic information, and low-level feature maps containing texture information. This way, we can reserve enough image information to reconstruct the industrial products with local and detail information.

The data-sampling strategy is another significant part of training neural networks. Previous implicit-field-based reconstruction methods neglected it and used a default sampling strategy to train their networks. However, fitting the decision surface is still a challenging task, especially in those implicit-field-based networks with complex branches. If the training data is hard to learn at the beginning, it will take a lot of time to converge. However, if the training data is too simple to indicate the exact surface of products, it will hurt the accuracy of the reconstruction in the end. The existing methods have to face this problem because the same sampled data is used throughout the training process. It is difficult for them to adaptively find a balance between the training points. Therefore, we propose an adaptive data-sampling strategy that can dynamically change the learning difficulty by varying input points according to the status of the network. It turns out that this design is able to accelerate the training speed and also improve the performance of the final fine-tuned network. In our paper, we aim to build a connection between physical and virtual spaces using single-view reconstruction technology for industrial products. We reconstruct 3D models in a digital twin field with a convenient and robust image input instead of complex high-cost multi-sensors. However, the existing single-view reconstruction methods cannot perform well on detail reconstruction. We propose a new feature-extractor network and a novel adaptive data-sampling strategy to reconstruct the 3D models together with detail and edge optimization. Then, we perform comprehensive experiments on both ShapeNet and a collected industrial dataset, the results of which indicate that our proposed method can achieve better performance than other SOTAs. The main contributions of this paper are summarized as follows:We leverage SVR technology to model physical industrial products in digital space using only one image. Experiments on industrial datasets show that implicit-field-based reconstruction methods have great potential for the digital twin field.We propose a feature extractor network for implicit field learning. An extra network is used to take advantage of FPN architecture that combines multi-level feature maps to extract feature vectors with rich semantics and texture information.A novel adaptive data-sampling strategy is proposed to accelerate and stabilize the training of the network and improve the performance of the reconstruction.

The rest of the article is organized as follows: Section 2 summarizes the related work. Section 3 provides definitions of symbols for implicit field learning and describes their usage in DT. In Section 4, the proposed network architecture and data-sampling strategy are introduced. Then, experimental comparisons are provided in Section 5. At last, Section 6 concludes this article.

## 2. Related Work

### 2.1. Digital Twin Technology

The digital twin represents a digital system of natural objects or subjects, including their data, function, and communication capabilities [1]. Digital representation brings convenience for analysis, optimization, verification, and validation, and therefore it can reduce the cost of industrial production. Simulation technology is key to the application of DT. The authors of [17] claim that machine tools can be simulated as virtual machine tools in a safe and cost-effective way. They propose an integration of manufacturing data and sensory data for developing a digital twin of machine tools to improve accountability and capabilities for cyber-physical manufacturing. A similar view is presented by the authors of [18], who emphasize the importance of simulation more. They propose an experimental digital twin (EDT), which means the application of simulation techniques brings digital twins to life and makes them experimental. In addition to virtual and physical industrial equipment, [19] also takes a human into consideration and presents a human-cyber-physical system in intelligent manufacturing. They also consider that the new generation of digital manufacturing consists of three main factors: human, network, and physical system.

Another critical challenge is digitalization technology, which constructs and maintains the digital twin of the existing physical system. Based on deep learning technology, [20] proposes a hybrid neural network model and a small object detection algorithm. Then, a cyber-physical system is built, with the aim of realizing dynamic synchronization between a physical manufacturing system and its virtual representation. The concern of [21] is the poor flexibility of current intelligent manufacturing systems caused by their centralized architecture. They propose applying blockchain technology to build a decentralized industrial Internet of things. The same idea is shared by [22], which presents an iterative bi-level hybrid intelligence model combining a permission blockchain with a holistic optimization model. They use smart contracts to decentralize task execution among machine tools and a digital twin model to apply coarse-grained holistic optimization.

### 2.2. Implicit-Field-Based SVR Methods

The existing SVR methods can be classified into two categories: geometry-based and implicit-field-based methods. The geometry-based methods can reconstruct 3D objects by generating visible geometry representations, such as meshes [23,24], voxels [25,26], and points [27,28]. However, meshed points are sparse and irregular, which makes them difficult to analyze. The storage occupied by a voxel is cubic of its resolution, so we would face a difficult balance between the quality and the storage cost of the voxels.

Implicit field learning avoids the above problems by representing the 3D shape implicitly and has brought significant improvement for SVR. A similar idea is presented in [10,11,12], which make predictions for each 3D point to reconstruct 3D shapes. This design avoids complex geometric representations of 3D shapes and is beneficial for producing contiguous surfaces. However, due to the flaws in the perception of objects in these works, the reconstructed shapes are too coarse [13]. Some critical parts of shapes, such as edges and corners, are disconnected or linked up at the wrong scale. To address this problem, refs. [14,15] extract feature vectors, combining global and local 2D features. Furthermore, the work in [29] independently extracts the hierarchy of local elements at different levels and performs recombination of these partial shapes in various sizes. These methods can capture richer local information from the input image and perform a more accurate reconstruction. However, these methods have to introduce a complex architecture and coarse results, which make them hardly expanded or embedded. In our paper, we avoid this problem by simplifying the camera assumption and presenting a concise reconstruction pipeline.

### 2.3. Detail Reconstruction with SVR

Compared to multi-view reconstruction (MVR), there is much less useful visual information in SVR. Though SVR methods can reconstruct 3D shapes successfully, some details are usually neglected. With the development of SVR, there has been a tendency towards reconstruction focusing on details. Local feature encoding is a common idea to address this problem. In [30], an hourglass network is used to extract feature maps from input images. Then, 3D points are mapped onto the 2D feature maps to extract the point-wise feature vector. In [15], 3D shape reconstruction is decoupled into shape reconstruction and residual reconstruction. The former produces a smooth main body according to the global feature vector, and the latter generates the residual shape to reconstruct the details. Aiming to learn the 3D shape from a global perspective, the result is projected onto a 2D plane, and the difference between the projection and the ground-truth shape is used as a part of the loss function. Besides local feature encoding, many other technologies have been tried. For example, [31] proposes a minimum circumference loss that trains the network in an easy-to-hard way. At the beginning of training, the loss function has a high tolerance, and the network focuses on the reconstruction of the main body. Then, the penalty for false prediction is increased, which helps supervise the learning of detail reconstruction.

## 3. Overview of Implicit Field Learning

### 3.1. Problem Definition

Given an image *I* of a product, the goal of SVR is to reconstruct a 3D digital shape *O* that captures not only the overall structure but also fine-grained details of the product. Implicit-field-based methods use an implicit function *f* to represent the digital shape.
(1)$$f\left(p,z\right)=\left\{\begin{array}{cc} 1, & \mathrm{if}\;p\;\mathrm{is}\;\mathrm{occupied}\;\mathrm{by}\;O,\\ 0, & \mathrm{otherwise}. \end{array}\right.$$
where *p* is any point in 3D digital space, and *z* is called the feature vector and represents the reconstructed object.

The task of implicit field learning is to train a feature extractor network *E* and a reconstruction network *D*. Network *E* is trained to extract feature vector *z*, and network *D* is designed to fit *f*.
(2)$$\begin{array}{cc} z & =E(I)\\ D(p,z) & =\left\{\begin{array}{cc} 1, & \mathrm{if}\;p\;\mathrm{is}\;\mathrm{occupied}\;\mathrm{by}\;O,\\ 0, & \mathrm{otherwise}. \end{array}\right. \end{array}$$

To train the networks, we need to sample a number of point–value pairs (p, v) as training data from the ground-truth shape *O*, where *v* is the label indicating whether the point *p* is occupied by *O* or not. Given a batch of point–value pairs (P, V) and the corresponding image *I*, we use the binary cross-entropy (BCE) function as the loss function and formulate it as follows:(3)$$\begin{array}{cc} L(I,P,V) & =\sum_{p\in P,v\in V}v\log v^{\prime }+\left(1-v\right)\log\left(1-v^{\prime }\right)\\ v^{\prime } & =D(p,\;z) \end{array}$$

Then, gradient descend algorithms, such as the Adam optimizer [32], can update the parameters of *D* and *E*.

### 3.2. Overview of the SVR Pipeline

Figure 1 shows the network architecture of common implicit-field-based SVR methods. It consists of feature extractor network *E* and reconstruction network *D*. The network *E* is usually implemented by CNN and is responsible for extracting the feature vector *z* from the input image *I*. The reconstruction network *E*, typically a multi-layer perceptron (MLP), takes the feature vector *z* and a 3D point *p* as input and predicts the label *v* of the input point. To generate surfaces from the implicit field in the inference phase, we can construct a 3D grid of the specified resolution, where 256×256×256 is commonly used. Then, the fine-tuned networks are used to make a prediction for each grid point so that the 3D grid contains the voxelized reconstruction result. Finally, the marching cube algorithm [33] is applied to extract the surface from the voxelized shape. Thus, we can get various geometric representations from implicit field learning.

## 4. Method

Based on the existing SVR methods, we propose two improvements. First, we propose an FPN-based feature extractor network to obtain rich information in the feature vector. Figure 2 shows the overview of our network architecture. Then, we design a novel adaptive data sampling strategy to get point–value pair (p, v) to train the networks efficiently and stably.

### 4.1. Feature Extractor Network

In implicit field learning, *z* is the only basis of network *E* for learning about the reconstructed product. It is natural that one feature vector *z* represents one product. Figure 3a shows the ordinary feature extraction, where the final feature map from CNN is average-pooled into a feature vector *z*. However, it also could be a bottleneck of the reconstruction algorithm. To alleviate the burden on *z*, we can change the form of network *E* as follows, which means one feature vector represents just one point of the product
(4)z=E(I, p)

This can be implemented as shown in Figure 3b. Several feature vectors of the point *p* are extracted from different levels’ feature maps and then concatenated into the final feature vector *z*. In this architecture, feature maps from different levels are leveraged. However, the feature maps from low levels are obtained by a few convolutional layers and so contain limited semantic information.

To tackle the above problems, we propose the feature extractor network shown in Figure 3c. The backbone network consists of four stages, the same as common CNN designs, and so will produce four feature maps from different levels. The low-level feature map is obtained by a few layers of convolutions and contains rich texture information but little semantic information. To address this problem, an extra network is used to combine the feature map from different levels in reverse order. Then, four combined feature maps are produced, and we can get *z* corresponding to the point *p* from them.

To gather *z* from 2D feature maps, the 3D point $p(p_{x},\;p_{y},\;p_{z})$ in space should be mapped onto 2D point $p^{\prime }\left(p_{x}^{\prime },\;p_{y}^{\prime }\right)$ in the image plane, where *x* and *y* are axes in the 2D image plane and *z* is the axis perpendicular to them. The commonly used method [14,15] uses a camera pose estimation network to predict the parameters of the projective transformation, which leads to more computations and errors. We notice that when the distances between each part of the product and the camera do not change too much, this transformation can be treated as an orthographic projection. This is usually the case with industrial products. Therefore, the 2D point $p^{\prime }$ can be computed as follows:(5)$$p^{\prime }=\left(p_{x}^{\prime },\;p_{y}^{\prime }\right)=\left(a\cdot p_{x},\;b\cdot p_{y}\right)$$
where *a* and *b* are parameters representing the ratio between the product sizes in the 2D image and 3D space and can be computed easily. Then, we can locate $p^{\prime }$ in the feature maps and apply a bilinear interpolation algorithm to get the corresponding feature vector. The feature vectors from four feature maps are concatenated into the final feature vector *z*.

### 4.2. Adaptive Data Sampling Strategy

As mentioned in Section 3, the training data of the networks is in the form of point–value pairs (p, v). The value v∈{0, 1} indicates whether the point *p* is occupied by the product or not. The binary labels carry limited information and lead to inefficient training. We notice that properties of the function y=slogx are useful to improve binary classification learning. Firstly, *y* is a strict convex function of *x*. This property can be leveraged to produce different weights for training different samples. Secondly, *s* is a hyperparameter controlling a function that is closer to a linear function or a piece-wise function.

To apply the function to implicit field learning, we redefine *v* as follows:(6)v(d, s)=−slog(m·d+n)
where *d* denotes the distance between the point *p* and the product surface, while *m* and *n* are constant values and can be solved according to the following constraints:(7)$$\forall s\in \left(0,\;1\right),\left\{\begin{array}{cc} v(0,\;s) & =1\\ v(d^{*},\;s) & =0 \end{array}\right..$$

The first constraint means when the point *p* is on the product surface (d=0), the label *v* should be a positive label. The second one means when the point *p* is far away from the surface ($d\geq d^{*}$), the label *v* should be a negative label. In addition, $d^{*}=\sqrt{5^{3}}$ is a hyperparameter representing that we only care about the 5×5×5 cube neighborhood of *p*. Then, we can solve the following:(8)$$\left\{\begin{array}{cc} m & =\frac{1-e^{-\frac{1}{s}}}{d^{*}}\\ n & =e^{-\frac{1}{s}} \end{array}\right.$$

Correspondingly, *v* can be defined as follows:(9)$$v\left(d,\;s\right)=\left\{\begin{array}{cc} -s\log(\frac{1-e^{-\frac{1}{s}}}{d^{*}}d+e^{-\frac{1}{s}}) & \mathrm{if}\;d\leq d^{*},\\ 0 & \mathrm{otherwise}. \end{array}\right.$$
where *s* is a value indicating the training status of the networks (the smaller, the better). As shown in Figure 4a, *v* is negatively correlated with *d*, which means the closer a point is to the surface, the more we expect the network to predict it as an occupied point. In addition, *v* is a strict convex function of *d*. This property will produce a much larger penalty for points closer to the surface and a relatively smaller penalty for farther ones. This mechanism can also be understood as adding a larger weight to hard samples and will help the networks to learn more efficiently.

The new definition of *v* changes it from a binary label to a contiguous value, which makes it so that each (p, v) can reveal more information about the surface. This way, the learning difficulty is reduced at the beginning of training. However, it will be hard for the networks to fit the exact surface, because *v* will be ambiguous if the point *p* is close to the surface, and so the networks will only reconstruct a rough and bloated shape. To tackle this problem, we set *s* dynamically in the training process. As shown in Figure 4b, when *s* is close to 1, the curve is smooth and descends slowly. However, when *s* is small (e.g., 0.1), the curve becomes steep, and *v* is close to the binary label. Based on this phenomenon, we can claim that the value of *s* controls the learning difficulty. Intuitively, at the beginning of the training process, *s* should be assigned a relatively large value. The training samples’ (p, v)s contain information about *p*’s neighborhood. Therefore, the networks can learn efficiently and arrive at a stable status. Then, *s* should be set larger so that the networks can fit the product’s exact surface.

To avoid setting *s* manually in the training process, we choose *F*1-*score* [34] to evaluate the training status and assign *s* as follows:(10)s=1−λ·f1_score
where λ=1.5 is used to adjust the value to an appropriate range. We call point–value pairs (p, 1) positive samples and the others negative ones. We also use tp (true positive) and fn (false negative) to denote the number of positive samples that the networks predict correctly and incorrectly, respectively, while fp (false positive) denotes that of negative samples predicted incorrectly by the networks. Therefore, the *F*1-*score* can be defined as follows:(11)$$\begin{array}{cc} precision & =\frac{tp}{tp+fp}\\ recall & =\frac{tp}{tp+fn}\\ f1\_score & =2\frac{precision\cdot recall}{precision+recall} \end{array}$$

We compute the *F*1-*score* every five epochs and assign it to *s* if it is larger than the previous score. At the beginning of training, we compute the ratio *r* of positive samples over all samples and assume that the networks perform prediction randomly. Therefore, the initial *s* is computed as follows:(12)$$\begin{array}{cc} precision & =r\\ recall & =\frac{1}{2}\\ f1\_score & =2\frac{r\cdot \frac{1}{2}}{r+\frac{1}{2}}=\frac{2r}{1+2r} \end{array}$$

## 5. Experiments

To evaluate the proposed method, we first compare it with the state-of-the-art algorithms on a general 3D shapes dataset [35]. Then, we conduct experiments on industrial products to prove the practicability of our SVR technology for DT.

### 5.1. Implementation Details

To implement our method, ResNet-18 [36] is used as the backbone network, and the implementation of FPN is followed [16]. The implicit field reconstruction network consists of five linear layers, each followed by a Leaky-ReLU [37] activation and a batch normalization [38] layer. To sample the training point–value pairs, we first sample 9872 points from the surface of objects and add random offsets to the points. We also generate 128 random points in the 3D spaces. Then, we assign values for the 10,000 points by the strategy proposed in Section 4.2 and obtain the training point–value pairs. We use the binary cross-entropy function as the loss function and Adam optimizer with learning rate $1e^{-4}$. We train the networks for 50 epochs. For training stability, we adjust the value of *s* every 10 epoch instead of every epoch. At the inference phase, the threshold of the marching cube algorithm is 0.5.

### 5.2. Dataset and Metrics

For comparison with other SVR methods, we conduct experiments on 13 categories of the ShapeNet [35] dataset, including 43,781 objects. The input images are provided by 3D-R2N2 [25]. As is common, 23 images per object are used in training, and the last image is used for testing. The voxel dataset provided by HSP [39] is used for training the data sampling. We follow the same train/test split as IM-NET.

The quantitative metrics we used are intersection of union (IoU), Chamfer-$L_{1}$ distance (CD), edge Chamfer distance (ECD), and DR-KFS. Edge Chamfer distance [10] is computed in the same way as Chamfer-$L_{1}$ distance, but only takes edge points into consideration. For a point *p*, we define its sharpness σ(p) as follows:(13)$$\sigma \left(p\right)=\min_{p^{\prime }\in N_{\epsilon }\left(p\right)}\left|n_{p}\cdot n_{p}^{\prime }\right|$$
where $N_{\epsilon }\left(p\right)$ denotes neighbor points of *p* within distance ϵ, while $n_{p}$ and $n_{p}^{\prime }$ are the unit normal vectors of *p* and $p^{\prime }$. We set ϵ to 0.01, and only points with σ(p)<0.1 are treated as edge points. DR-KFS is implemented according to [40], and the results are normalized into [0, 1].

### 5.3. Experiments on General Objects

Our method is compared with Pixel2Mesh [24], AtlasNet [41], OccNet [10], IM-Net [11], and DISN [14]. OccNet and IM-Net both use the most common network architecture, shown in Figure 1. OccNet trains the whole pipeline in a end-to-end way, while IM-Net trains the CNN and MLP separately. Based on OccNet, DISN improves the feature extraction network, as shown in Figure 3b. All three methods use binary labels to train the networks.

**Qualitative Results.**Figure 5 shows the qualitative results. OccNet is more likely to produce thick meshes. This is a benefit in generating the main body of the 3D object but harmful to reconstruction of the details and edges. For example, OccNet can reconstruct the body of a handgun, but fails to recover the trigger. In contrast, IM-Net tends to produce thin meshes, but also generates more fragments. As we can see, it cannot reconstruct the complete connections for chair legs and airplane engines. DISN makes accurate reconstructions for some categories, such as chairs and tables. However, occasionally it cannot generate complete planes and fails to generate flat edges. The results of our methods are shown in the last column. Not only the main body and structure of 3D objects are reconstructed, but more details are also reserved.

**Quantitative Results.**Table 1 shows a quantitative comparison of the seven categories with the most shapes in the ShapNet-Core dataset [35]. The mean value of each metric in computed over all samples. The ↓ means the lower, the better; while ↑ is the opposite. As shown in Table 1, our method shows the best performance on most categories. Especially on chairs and tables, our method significantly outperforms the other methods. As mentioned above, OccNet does well in the reconstruction of objects’ main body, but not in generating edges and details. The main body of an airplane or a car takes a large part of the whole shape, so the OccNet has outstanding results. However, over all the categories of ShapeNet-Core, our method shows a better performance.

**Ablation Study.** We conduct ablation studies on the proposed design, and the results are shown in Table 2. The first row describes the used feature extractor network. *CNN* represents when only the ordinary CNN, ResNet-18, is used. *CNN + FPN* represents the proposed design where the extra FPN is also applied. The second row describes which data sampling method is used; *binary* means *v* is assigned a binary label, as described in Section 3. In addition, v(d, 0.5) means that the proposed function v(d, s) is applied but *s* is fixed to 0.5, so that the learning difficulty can be dynamically adjusted, while v(d, s) denotes the adaptive data-sampling strategy. It turns out that the application of FPN can bring significant improvements over ordinary CNN. Therefore, we claim that the simple feature vector mechanism can severely limit the performance of SVR methods. The value function v(d, s) is also effective. Compared with binary values, our v(d, s) design can carry more information and apply an appropriate penalty on the false predictions.

Furthermore, to clarify the influence of v(d, s) on the reconstruction results, we fix the *s* to different values through the training phase and repeat the network training. The qualitative results are shown in Figure 6. When *s* is fixed to 0.9 or 0.7, the network produces bloated meshes. The main shape is reconstructed, but most details are lost. When *s* is set to a small value, such as 0.1, the network generates slim meshes. Sometimes the reconstruction results are delicate, but occasionally the surfaces are broken, and even the main shapes cannot be reconstructed. According to Figure 4a, this makes sense for the results shown in Figure 6. Even though the points are far away from objects’ surface, as long as they are in the neighborhood controlled by $d^{*}$, they can get a positive value. This is quite different from the binary labels. When *s* is set to a large value, such as 0.9 or 0.7, the points can be assigned to relatively large values. After training with these point–value pairs, the network is more likely to predict input points as positive. With these predictions, the bloated meshes are generated by the marching cubes algorithm. When *s* is set to a small value, the predictions tend to be negative, and slim meshes are produced. Therefore, it is intuitive that the proposed adaptive data-sampling strategy works by choosing the appropriate value of *s* dynamically according to the training status.

### 5.4. Experiments on Industrial Machines

We collect 400 3D models of different categories and their 2D rendered images on public 3D model websites, such as TurboSquid, SketchFab, and cgmodel, and paid 3D models on Taobao. We divide the 3D models into four categories, including 195 industrial machines, 138 industrial components, 45 metaverse buildings, and 22 cartoon characters. Here, we use seven out of ten for training, two out of ten for testing, and one out of ten for validation in each category.

Then, we apply our SVR method to prove the practicability of this technology. Unlike the general 3D objects dataset, the number of industrial machines is relatively small, but each machine has a more complex structure and richer details. However, there are not enough models in the categories of metaverse buildings and cartoon characters. Therefore, we only show the qualitative reconstruction results of industrial components and machines in Figure 7. The sophisticated components and machines can be fully reconstructed, but the main structures are basically generated. Considering the reconstruction results, we believe the SVR technology has its own place in the field of industrial DT.

We also compare our methods with Pixel2Mesh, OccNet, IM-NET, and DISN on our collected data to show the effectiveness of our methods quantitatively in Table 3. We evaluate each method with ECD and DR-KFS. ECD can leverage the edge and detail reconstruction quality, and DR-KFS can provide a global view of reconstructed shapes. It is obvious that our method outperforms the other methods on our collected data. In our collected data, there are rich details and edge information with hard topology. Using our methods can optimize the detail and edge reconstruction, which makes our methods perform better than others.

## 6. Conclusions

In this paper, we first introduce single-view reconstruction technology to build a connection from the physical space to the digital space. Through SVR technology, we can generate the digital twins of industrial productions just by relying on a single image. Compared with complex multi-sensor-based reconstruction, the convenience and cheapness of 2D images can be leveraged in the intelligence industry. It is still challenging for SVR to be utilized in the industrial world, because even if the existing SVR methods can reconstruct the shapes, they all suffer from over-smooth surface and a lack of details and edges.

To address this problem, we propose a feature extractor network and a novel data-sampling strategy in our paper. We first use CNN to extract feature maps instead of a single feature vector. Inspired by FPN, we also design an extra convolutional network to combine high-level feature maps and low-level feature maps. Then, we use a billinear interpolation algorithm to extract the feature vector corresponding to the input point. We also design an adaptive point–value (p, v) pairs sampling strategy. This strategy treats *v* as the function v(d, s). This way, each point–value pair can carry more information about the surface. By setting *s* to different values according to the training status, the learning difficulty can be automatically adjusted.

For the methods proposed in this paper, there are still some improvements that can be made. For example, when we extract the feature vector of the point, we simplify the perspective transformation, which limits the usage scenario and may be designed in a more elegant way. Furthermore, the technology of the model ensemble is also a direction worth trying.

In the future, we will explore the SVR technology for industrial DT deeper, aiming to improve the quality of the reconstructed digital model. Besides, a larger 3D industrial dataset should be collected and reorganized so that more research on 3D reconstruction for DT can be conducted.

## Figures and Tables

**Figure 1 sensors-22-06630-f001:**
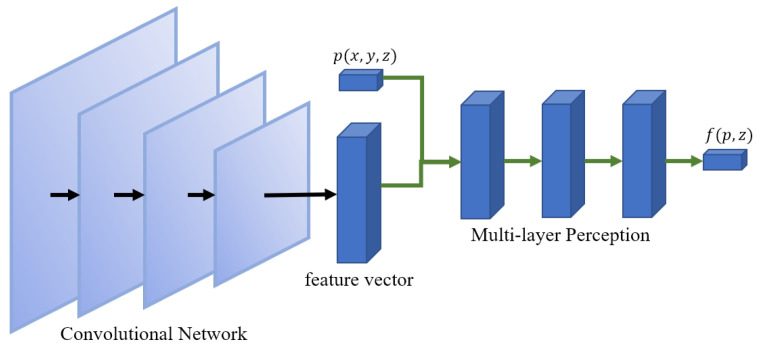
Common network architecture of implicit-field-based SVR.

**Figure 2 sensors-22-06630-f002:**
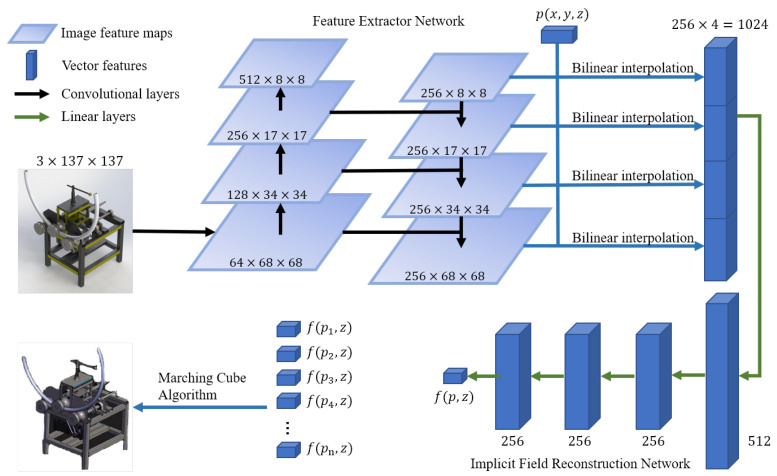
Pipeline of our proposed method.

**Figure 3 sensors-22-06630-f003:**
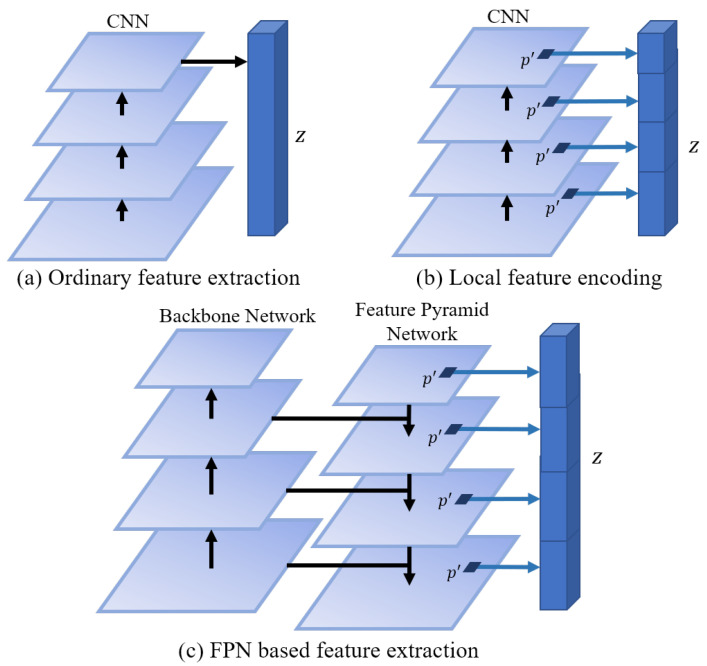
Different structures of feature extractors.

**Figure 4 sensors-22-06630-f004:**
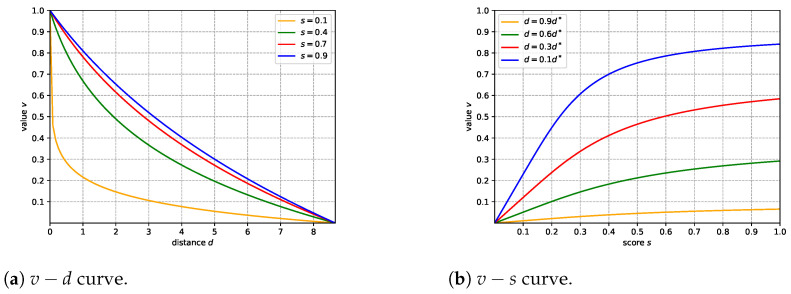
Visualization of v(d, s).

**Figure 5 sensors-22-06630-f005:**
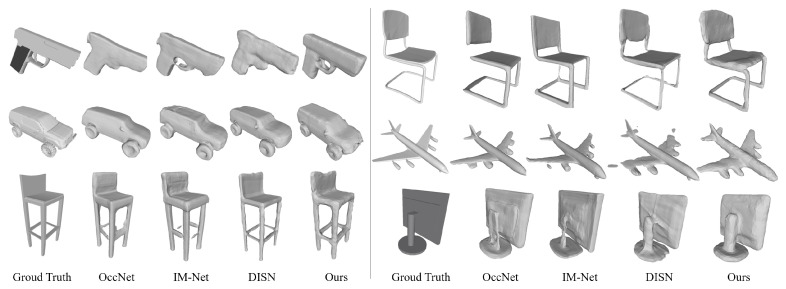
Reconstruction results on industrial machines.

**Figure 6 sensors-22-06630-f006:**
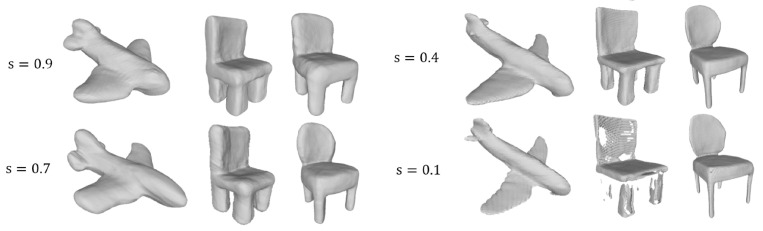
Reconstruction results trained with *s* different values.

**Figure 7 sensors-22-06630-f007:**
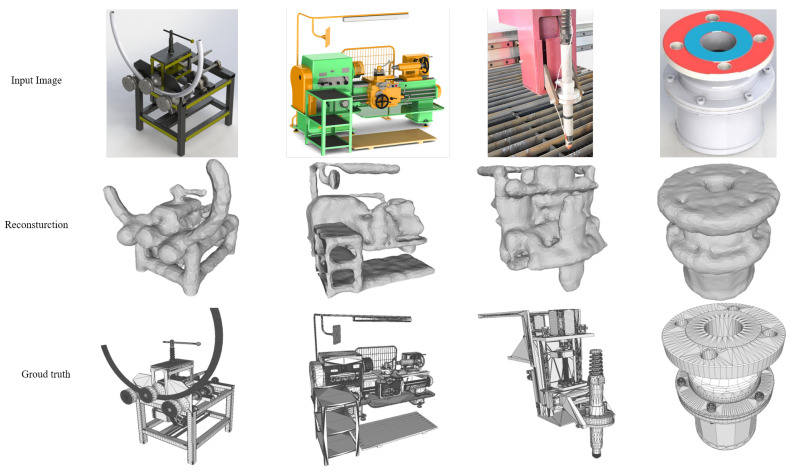
Reconstruction results on industrial machines.

**Table 1 sensors-22-06630-t001:** Quantitative comparison between different SVR methods. Bold means the best result in current table item.

	Method	Airplane	Car	Chair	Display	Lamp	Rifle	Table	Mean
IOU(↑)	Pixel2Mesh	0.423	0.524	0.311	0.475	0.238	0.429	0.408	0.401
	AtlasNet	0.451	0.535	0.366	0.480	0.217	0.455	0.430	0.419
	OccNET	**0.480**	0.570	0.358	0.439	0.254	0.427	0.461	0.461
	IM-NET	0.379	0.674	0.487	0.514	**0.336**	0.468	0.484	0.527
	DISN	0.328	0.672	0.301	0.358	0.189	0.197	0.105	0.360
	Ours	0.443	**0.710**	**0.532**	**0.552**	0.295	**0.671**	**0.505**	**0.592**
CD(↓)	Pixel2Mesh	0.587	0.414	0.662	0.641	0.702	0.521	0.796	0.617
	AtlasNet	0.592	0.440	0.651	0.632	0.695	0.438	0.701	0.593
	OccNET	**0.461**	0.368	0.639	0.636	0.683	0.414	0.763	0.587
	IM-NET	0.574	0.650	0.919	0.907	0.802	0.556	0.979	0.797
	DISN	0.572	0.645	0.907	0.906	0.800	0.578	0.972	0.794
	Ours	0.562	**0.344**	**0.597**	**0.613**	**0.637**	**0.325**	**0.653**	**0.552**
ECD(↓)	Pixel2Mesh	0.565	0.477	0.589	0.582	0.677	0.430	0.742	0.580
	AtlasNet	0.601	0.397	0.506	0.658	0.679	0.426	0.688	0.565
	OccNET	**0.423**	**0.288**	0.465	0.475	0.581	0.321	0.594	0.473
	IM-NET	0.522	0.370	0.627	0.641	0.695	0.478	0.750	0.589
	DISN	0.554	0.412	0.732	0.653	0.733	0.565	0.844	0.655
	Ours	0.447	0.313	**0.385**	**0.469**	**0.547**	**0.296**	**0.536**	**0.452**
DR-KFS(↓)	Pixel2Mesh	0.338	0.490	0.551	0.568	0.592	0.487	0.605	0.519
	AtlasNet	0.299	0.401	0.499	0.524	0.559	0.463	0.592	0.425
	OccNET	**0.296**	**0.239**	0.325	0.366	0.402	0.291	0.438	0.337
	IM-NET	0.337	0.308	0.375	0.386	0.398	0.269	0.512	0.367
	DISN	0.324	0.313	0.392	0.403	0.422	0.401	0.524	0.397
	Ours	0.320	0.274	**0.314**	**0.361**	**0.374**	**0.258**	**0.373**	**0.288**

**Table 2 sensors-22-06630-t002:** Quantitative results of ablation study.

	CNN	CNN + FPN	CNN + FPN	CNN + FPN
	Binary	Binary	v(d,0.5)	v(d,s)
IOU	0.524	0.553	0.567	0.592
CD	0.813	0.664	0.560	0.552
ECD	0.576	0.538	0.457	0.452

**Table 3 sensors-22-06630-t003:** Quantitative comparison between different SVR methods on our collected data. The bold means the best result when comparing with others.

	Method	Machines	Components	Buildings	Cartoons	Mean
ECD(↓)	Pixel2Mesh	0.828	0.801	0.874	0.853	0.839
	AtlasNet	0.802	0.783	0.859	0.848	0.823
	OccNET	0.795	0.763	0.848	0.820	0.806
	IM-NET	0.789	0.771	0.852	0.810	0.805
	DISN	0.798	0.784	0.851	0.816	0.812
	Ours	**0.724**	**0.712**	**0.801**	**0.790**	**0.757**
DR-KFS(↓)	Pixel2Mesh	0.742	0.709	0.885	0.838	0.793
	AtlasNet	0.757	0.701	0.873	0.850	0.795
	OccNET	0.695	0.673	0.794	0.771	0.733
	IM-NET	0.662	0.649	0.781	0.744	0.709
	DISN	0.678	0.641	0.790	0.724	0.724
	Ours	**0.589**	**0.576**	**0.703**	**0.699**	**0.641**

## Data Availability

The data that support the findings of this study are openly available in the following links: https://shapenet.org/, https://ai.taobao.com/ and https://www.turbosquid.com/, accessed on 22 July 2022.

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
