# Peer review of "Adaptive Points Sampling for Implicit Field Reconstruction of Industrial Digital Twin"

_sensors, 2022, doi:10.3390/s22176630_

Round 1

Reviewer 1 Report

Authors propose a feature extractor network based on Feature Pyramid Network for better details reconstruction trough SVR technology. In this perspective, an adapted network combining convolutional feature maps from different levels has been proposed.  

The idea has some novelty, but it deserves to be better presented. The major problem is associated with the writing methodology and the way the paper originality is presented. It is also difficult to classify the authors' contribution compared to existing works. 

For example, the adaptive data sampling strategy section seems to be the most important part of the paper. Unfortunately no formal description is introduced. The proposed strategy must be presented in an algorithmic way. This main contribution must have a debatable theoretical background that shows its robustness and the possibility to extend it. 

The article is terribly lacking in illustration. 

The bibliography needs to be improved and updated. Here are some suggestions (Please respect the MDPI format when formatting; the year of publication of an article must be mentioned)

doi : 10.3390/jmmp5030080

doi: 10.1109/DTPI52967.2021.9540133.

doi: 10.1109/ITNEC48623.2020.9084652.

doi: 10.1109/DTPI52967.2021.9540131.

doi: 10.1109/DTPI52967.2021.9540090.

doi: 10.1109/DTPI52967.2021.9540078.

I suggest editing to improve the readability of the paper. It is recommended that author proofread  paper and improve flow and cohesion. The writing style and language are expected to be at high standards.

Reviewer 2 Report

The authors propose a method to "leverage SVR technology to model physical industrial products into digital space using only one image", defining a "feature extractor network for implicit field learning" and a "novel data sampling strategy to accelerate and stabilize the training of
network".

While the idea of the manuscript seems really interesting, I have some doubts about the goals set by the authors and the conclusions drawn.

First of all, one of the main pros of the employment of 3D models of products in DT of industrial environments is the monitoring of the quality of the objects under production and of potential issues that may arise in the collision between products and machines. However, the proposed results are of poor quality and unsuitable for the production environment, even if they improve on state of art in SVR.  This said, I don't understand why the authors conclude that "the convenience and cheapness of 2D images can be leveraged in intelligent industry", since the results are of such poor quality.

Second: the authors aim at generating 3D models of products because "they play important roles throughout different stages of industrial production, including product design, manufacturing and maintenance." However, all the proposed experiments are done on "general 3D shapes dataset" and on "industrial machines" and none is dedicated to a product.

Another issue regards the choice of 3D representation of shapes. The authors say that "meshes and points" reconstructed through SVR "are sparse and irregular" without detailing. As far as I can see, for example in "Pixel2mesh: Generating 3D mesh models from single rgb images." the quality of the reproduced shape is better that the results of this paper. The authors even say that "the storage occupied by a voxel is cubic of its resolution, so we would face a hard balance between quality and store cost of voxels.", however:

1) if the quality of the reconstructed (voxelized) shape is good, a surface-based representation can be obtained from it (e.g., meshes)

2) the proposed method reconstructs a voxelized shape and then extracts the implicit surface through Marching Cubes.

This said I do not understand the motivation of the authors.

Finally, the paper presents a huge amount of grammatical issues (e.g.,  line 30 - "But the connection between real products and 3D digital models is a challengechallenging problem") and typos (e.g., tickled instead of tackled).

Reviewer 3 Report

This paper proposes a feature extractor network for implicit field learning and a novel adaptive data sampling strategy to accelerate and stabilize the training. Specifically, the proposed method take advantage of the SVR technology to generate the digital twins of industrial products starting from a single image. The method is composed of two main ingredients: ResNet-18 (backbone network) and an implementation of FPN.

Though my experience on the topic is limited, I found the article interesting.

MAJOR COMMENTS:

-       I suggest a carefully review of the English language. The periods after “And” and “But” is not good.

-       The Introduction is very long and disorganized. I suggest you to divide it into subsections or to put some parts in Section 2.

-       At the end of Page 7, I suggest you to add the following citation
Max Kuhn, Kjell Johnson, Applied Predictive Modeling, Springer New York, 2018
for more details on the measures.

-       Section 5, Line 215 the references to the used datasets is missed.

-       In Table 2 the best results are not highlighted.

-       I suggest to do an appropriate subsection to better explain the measures used for the comparison.

-       The authors use ShapeNet dataset for the tests. Is it possible to consider other datasets?

MINOR COMMENTS:

-       The white space between a word and a parenthesis is missed (see for example the first line of the Abstract “twin(DT)”)

-       Page 6, Line 184: is consist consists

-       Page 9, Line 258: tab. 2 Tab. 2

-       Page 2, Line 79: An A

Round 2

Reviewer 1 Report

I genuinely appreciate the generous efforts of the given Authors to improve the quality of the paper, to push its scientific depth forward ,and to be vigilant towards its accuracy and consequently I accept  the Authors' statements and I show a lot of gratitude for that minute and detailed revision. 

However,  I still suggesting some editing to improve the readability of the paper. It is recommended that authors proofread the paper by native English speaker and improve flow and cohesion. I insist on the writing style and language. They are expected to be at high standards. 

Author Response

Responses to Reviewer 1: 

Thanks so much for you kind suggestions. We polished our paper in the latest version according to your advice. Also, we included more tables and figures to enrich our paper and make it easy to follow. 

--
Jiongchao Jin

Reviewer 2 Report

I appreciate the extensive correction of english flaws, however the points I have highlighted have not yet been satisfactorily answered. My comments can be found in the attached text.
